# Local GABA concentration is related to network-level resting functional connectivity

**Charlotte J Stagg[1]\*, Velicia Bachtiar[1], Ugwechi Amadi[1], Christel A Gudberg[1], Andrei S Ilie[1,2], Cassandra Sampaio-Baptista[2], Jacinta O'Shea[1], Mark Woolrich[1], Stephen M Smith[1], Nicola Filippini[1,3], Jamie Near[1], Heidi Johansen-Berg[1]**

[1]Oxford Centre for Functional MRI of the Brain (FMRIB), Nuffield Department of Clinical Neurosciences, University of Oxford, Oxford, United Kingdom; [2]Department of Pharmacology, University of Oxford, Oxford, United Kingdom; [3]Department of Psychiatry, University of Oxford, Oxford, United Kingdom

**Abstract** Anatomically plausible networks of functionally inter-connected regions have been reliably demonstrated at rest, although the neurochemical basis of these 'resting state networks' is not well understood. In this study, we combined magnetic resonance spectroscopy (MRS) and resting state fMRI and demonstrated an inverse relationship between levels of the inhibitory neurotransmitter GABA within the primary motor cortex (M1) and the strength of functional connectivity across the resting motor network. This relationship was both neurochemically and anatomically specific. We then went on to show that anodal transcranial direct current stimulation (tDCS), an intervention previously shown to decrease GABA levels within M1, increased resting motor network connectivity. We therefore suggest that network-level functional connectivity within the motor system is related to the degree of inhibition in M1, a major node within the motor network, a finding in line with converging evidence from both simulation and empirical studies.

**\*For correspondence:** charlotte. stagg@ndcn.ox.ac.uk

**Competing interests:** The authors declare that no competing interests exist.

## Introduction

There has been a surge of recent interest in so-called 'resting state networks' (RSNs) in the human brain. These robust, distributed networks, most commonly detected using functional MRI, show correlated fluctuations in their resting signal and have revealed networks formed from spatially widespread but anatomically and functionally closely linked regions (*Fox and Raichle, 2007*; *Snyder and Raichle, 2012*). Understanding the basis of these resting state networks is of increasing interest, as variation in the strength of functional coupling within these networks has been shown to be highly sensitive to a wide variety of clinical, genetic, or cognitive states (*Filippini et al., 2009*; *Pievani et al., 2011*). However, the neurochemical basis for such variations remains poorly understood.

Resting correlations have long been thought to be a functional marker of excitatory connections (*Vincent et al., 2007*; *Shmuel and Leopold, 2008*). However, recent simulation studies have predicted that patterns of local oscillatory activity can emerge spontaneously in a coherent fashion across large networks with specific connectional architectures (*Cabral et al., 2011*). Such local neuronal dynamics depend on neurochemical inhibitory tone, as shown, for example, by changes in local activity within the primary motor cortices (M1s) caused by pharmacological manipulations (*Hall et al., 2011*). If correlated activity spontaneously emerges through attractor dynamics in distant cortical regions, then extent of these functional correlations should be best predicted by local GABA concentrations.

**eLife digest** Even when your body is at rest, your brain remains active. Subjects lying in brain scanners without any specific task to perform show coordinated and reproducible patterns of brain activity. Areas of the brain with similar functions, such as those involved in vision or in movement, tend to increase or decrease their activity in sync, and these coordinated patterns are referred to as resting state networks.

The functions of these networks are unclear—they may support introspection, memory recall or planning for the future, or they may help to strengthen newly acquired skills by enabling the brain to replay previous learning episodes. There is evidence that resting state networks are altered in disorders such as Alzheimer's disease, autism and schizophrenia, but little is known about how these changes arise or what they might mean.

Now, Stagg et al. have used a type of brain scan called magnetic resonance spectroscopy to gain insights into the mechanisms by which one particular network—the resting motor network—is generated. This network consists of areas involved in planning, monitoring and executing movements, and includes the primary motor cortex, which initiates movements by sending instructions to the spinal cord.

The levels of a chemical called GABA—a neurotransmitter molecule that tends to inhibit the activity of nerve cells—were measured in the primary motor cortex of young healthy volunteers as they lay idle in a scanner. GABA levels were negatively correlated with the amount of coordinated activity within the resting motor network. By contrast, no relation was seen between coordinated activity and the levels of the neurotransmitter glutamate, which tends to increase the activity of nerve cells. Furthermore, when a weak electric current was applied through the subjects' scalp to their primary motor cortex—a technique previously shown to lower levels of GABA in the region— the resting motor network became stronger.

In addition to providing new information on how the rhythmic patterns of activity seen in the resting brain arise, the work of Stagg et al. contributes to the more general effort to understand the complex patterns of connections within the human brain.

Here, we carried out a series of experiments to address this hypothesis. In the first three studies, we directly quantified GABA within the left M1 using Magnetic Resonance Spectroscopy (MRS) to test the prediction that the strength of functional connectivity within the motor RSN is related to GABA concentration within M1, a major node of the network, and that this relationship is specific in both neurochemical and anatomical terms. In a fourth study, we performed an intervention study—to directly assess this relationship by testing whether application of anodal (excitatory) transcranial direct current stimulation (tDCS) to M1, an intervention known to decrease local GABA concentration (*Stagg et al., 2009*; *2011a*), resulted in strengthening of functional connectivity within the motor RSN.

## Results

MR Spectroscopy data were acquired from the hand region of left primary motor cortex (M1) and all neurochemicals of interest were expressed as a ratio to N-acetyl aspartate (NAA) (*Stagg et al., 2009*; see *Figure 1* for representative spectra).

We investigated resting functional connectivity using two different analysis approaches: independent component analysis (ICA) and seed-based correlation. For the ICA-based analysis approach, ICA was applied to group level resting fMRI data to define group level RSNs. Next, a standard dual-regression approach was used to regress these group level RSNs of interest against each individual subject's data. This generates subject level maps of functional connectivity for each RSN of interest. Analysis focused on the motor network, the principal RSN containing M1 (*Figure 2A*). For the seed-based correlation analysis, we correlated the average resting BOLD timecourses between the left and right M1s as a simpler measure of functional connectivity between left M1, from where our GABA measurements were acquired, and another major network node. To investigate the anatomical specificity of the relationship between GABA and functional connectivity within the motor RSN we correlated the average resting BOLD timecourses between the left M1 and left dorsal premotor cortex (PMd).

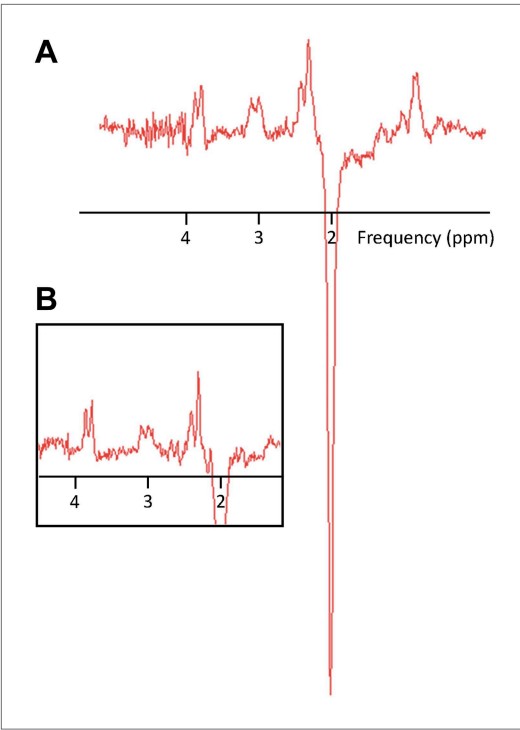

**Figure 1**. Representative MR spectra. (**A**) A subject with a high GABA:NAA ratio. (**B**) A subject with a low GABA:NAA ratio.

## Experiment 1

Experiment 1 considered a cohort of 12 young, healthy subjects, of whom 11 had spectra of sufficient quality for inclusion. Using ICA, we demonstrated a significant inverse correlation between functional connectivity within the motor RSN and GABA concentration within M1 (M1-GABA) in healthy young volunteers (r = −0.71, p=0.01; *Figure 2C*).

To test the anatomical specificity of this result, we assessed the default mode network (DMN), as this is a well-characterized RSN that does not include M1 (*Figure 2B*). There was no significant correlation between M1-GABA and functional connectivity within the DMN (r = 0.25, p=0.44) (*Figure 2D*), and M1-GABA and functional connectivity within the DMN was significantly less correlated than M1-GABA and functional connectivity within the motor RSN (Fisher's R-to-Z transformation, Z = −2.29, p=0.02).

To test the neurochemical specificity of this result, we measured M1 glutamate (assessed via Glx, a composite measure of glutamate and glutamine). There was no significant correlation between motor RSN functional connectivity and M1-Glx (r = −0.35, p=0.36; *Figure 2E*), and the correlation between M1-GABA and motor RSN functional connectivity persisted when M1-Glx was co-varied out (r = −0.67, p=0.03).

Additionally, using a seed-based correlation analysis, we demonstrated a significant inverse relationship between the degree of correlation between the two M1s and GABA concentration (r = −0.60, p=0.047; *Figure 3A*). There was a trend towards an inverse relationship between the degree of correlation between left M1 and left PMD and GABA concentration (r = −0.49, p=0.12; *Figure 3—figure supplement 1*).

## Experiments 2 and 3

We broadly replicated these findings in two further separate cohorts: once in a group of 16 young healthy subjects (experiment 2; see *Figure 2—figure supplement 1*) and one in a group of older adults (experiment 3; see *Figure 2—figure supplement 2*).

## Experiment 4

Experiment 4 aimed to investigate whether decreasing GABA within M1 using anodal tDCS (*Stagg et al., 2009*; *2011a*), increased functional connectivity within the motor RSN. We acquired resting fMRI data before and immediately after anodal tDCS applied to the left M1 in a separate group of 10 subjects. A significant increase in network functional connectivity after tDCS was observed in the motor RSN (pre: 20.7 ± 3.34, post: 26.7 ± 2.97; t(9) = 2.59, p=0.02; *Figure 4*). There was also a significant increase in the degree of M1-M1 correlation (t(9) = 1.94, p=0.04; *Figure 3B*).

## Discussion

In this study, we investigated the basis of the long-range fluctuations seen in resting fMRI. In three separate cohorts of subjects we have demonstrated a significant negative correlation between the levels of GABA in M1 and the strength of functional connectivity within the motor RSN. This relationship was specific both anatomically and neurochemically: no relationship between glutamate levels and resting connectivity was demonstrated. We believe that the fundamental relationship between resting connectivity within the motor network and M1 GABA demonstrated here is robust, as it is replicated across both ICA and seed-based approaches; whether the MRS and fMRI data

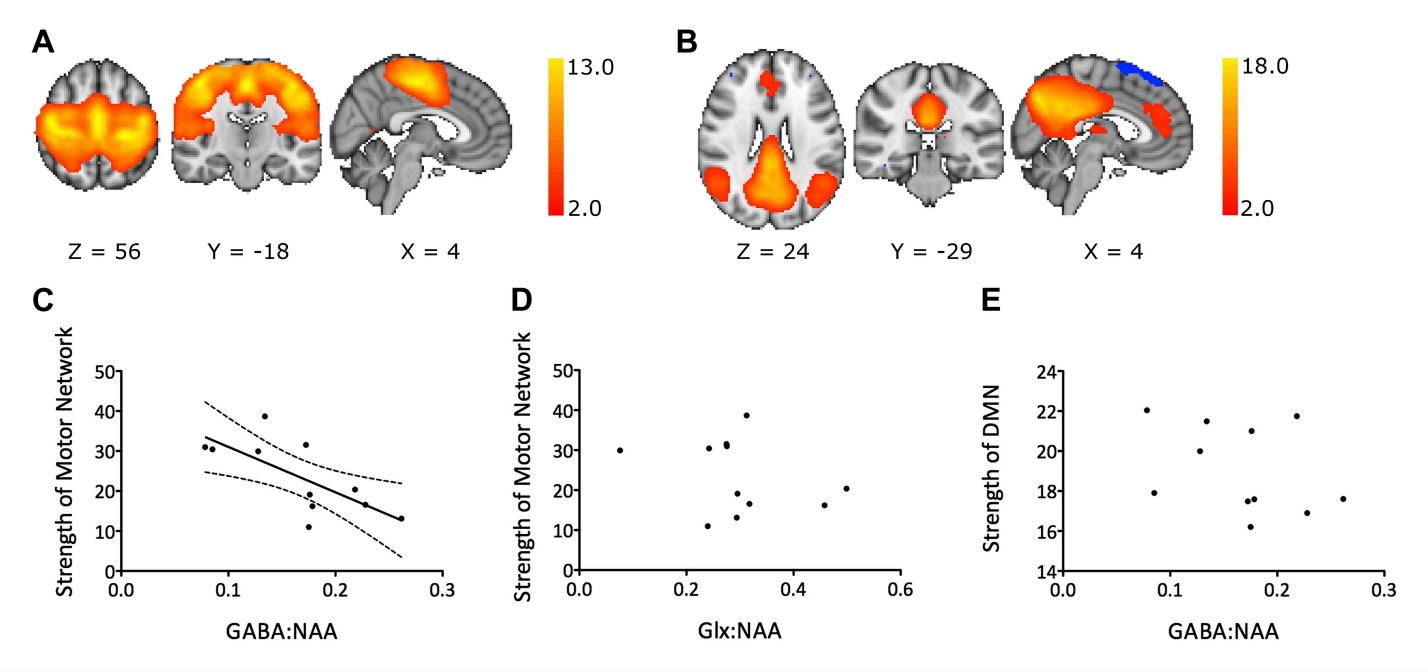

**Figure 2**. (**A**) Group mean motor resting state network. (**B**) Group mean default mode network. (**C**–**E**) Experiment 1: a significant relationship was demonstrated between M1-GABA and functional connectivity within the motor RSN (r = −0.71, p=0.01; **C**) but not between M1-Glx and motor network functional connectivity (**D**) nor between M1-GABA and functional connectivity within the DMN (**E**).

The following figure supplements are available for figure 2:

**Figure supplement 1**. Experiment 2 replicated the findings of experiment 1 in a separate group of 16 young, healthy subjects.

**Figure supplement 2**.

were collected on the same day or different days; across different scanners; and in young and old healthy cohorts.

Further, we have shown that a technique that is known to decrease GABA within M1 significantly increases functional connectivity of the motor RSN.

What do these findings tell us about how activity at a cellular level relates to long-range network connectivity? The activity within the motor RSN has been shown to be related to fluctuations in the power of beta oscillations (**Brookes et al., 2011**), which have in turn been related to GABA activity (**Hall et al., 2011**). A recent simulation-based paper suggested a positive relationship between local oscillatory power in the gamma band and resting state functional connectivity (**Cabral et al., 2011**), something supported in the visual network by a relationship between RSN connectivity and local gamma frequency oscillatory activity (**Shmuel and Leopold, 2008**). Evoked gamma power is also known to be inversely related to extra-synaptic GABA tone (**Towers et al., 2004**). GABA is present in the human brain in three major pools—in pre-synaptic vesicles, as a metabolite in the cytoplasm, and in the extracellular fluid, where it underlies extra-synaptic GABA tone. It is likely that MRS-assessed GABA, which is sensitive to the total *amount* of GABA within the voxel, more closely reflects extra-synaptic rather than synaptic GABA activity (**Stagg et al., 2011b**).

While it is not yet clear what the relationship is between local gamma oscillations and beta oscillations, the negative correlation found between motor RSN functional connectivity and MRS-assessed GABA in M1 is consistent with the idea that fluctuations in the power of oscillations underlie long-range resting functional connectivity.

In combining observational studies (Experiments 1–3) and an intervention study (Experiment 4), the data presented here significantly extends and brings coherence to previous findings. One previous study has observed a correlation between GABA in the posteromedial cortex and the strength of the

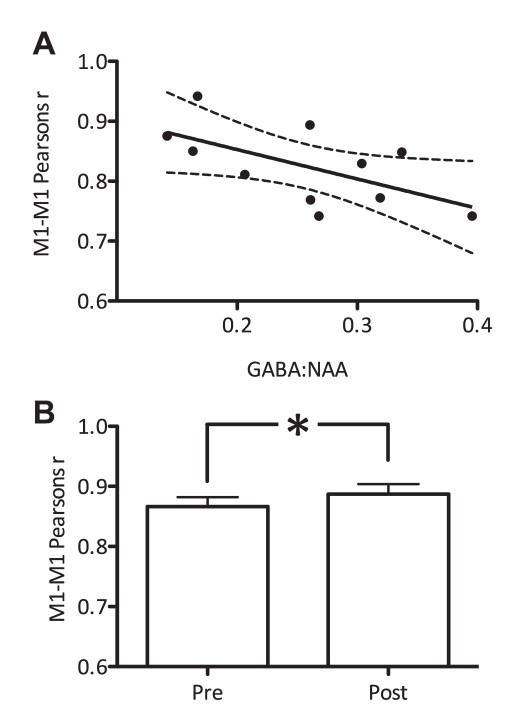

**Figure 3**. The degree of correlation between the left and right primary motor cortices (M1s) was significantly related to M1 GABA levels. Values shown are raw Pearson's correlation coefficients for ease of display. As these are not normally distributed all statistical analyses were performed on log-transformed data (see 'Materials and methods'). (**A**) Experiment 1 (r = −0.60, p=0.047). (**B**) Experiment 4: the correlation between left and right M1s was significantly increased after anodal tDCS (t(9) = 1.94, p=0.04).

The following figure supplements are available for figure 3:

**Figure supplement 1**. There was a trend towards a relationship between the degree of correlation between the left M1 and the left dorsal premotor cortex (PMd) and M1 GABA levels.

default mode network (*Kapogiannis et al., 2013*). Taken together with our findings, this suggests that the relationships we report are not specific to the motor system, but rather a general feature of resting state functional connectivity networks. It is interesting to note, however, that in addition to a relationship with GABA, Kapogiannis et al. demonstrate a significant positive correlation between glutamate in the posteromedial cortex and the strength of the default mode network. There are a number of reasons why we did not find a similar relationship between glutamate and resting state connectivity. Our MRS approach allowed quantification of Glx, a composite measure of glutamate and glutamine, and therefore may be less sensitive to relationships with glutamate specifically. Alternatively, it may be that different networks have different neurochemical profiles, although this seems less likely.

Other previous studies have explored the relationship between task-evoked BOLD responses and GABA: a relationship between GABA levels and BOLD signal in response to a task has been demonstrated in anatomically plausible regions for both motor and visual tasks (*Muthukumaraswamy et al., 2009*; *Stagg et al., 2011a*), such that higher local GABA concentrations are associated with smaller task-related BOLD signals. Motor-task related BOLD signals have also been shown to correlate with the strength of the motor RSN across individuals (*Kannurpatti et al., 2012*). The relationships demonstrated here, build on this prior work and provide a mechanistic explanation for the previous finding that motor learning, which decreases GABA (*Floyer-Lea et al., 2006*), increases RSN functional connectivity (*Albert et al., 2009*).

Taken together, these findings relate RSN functional connectivity to local inhibitory tone within the motor cortex, and thereby may reveal an important neural mechanistic explanation for the RSN changes observed in a wide range of states.

## Materials and methods

All subjects gave their informed consent to participate in the study in accordance with local ethics committee approval (Experiments 1, 3 & 4: Oxfordshire REC A 06/Q1604/2; Experiment 2: East London REC 1 10/H0703/50). All subjects were right-handed and none had a history of any neurological or psychiatric disorder.

### Experiment 1

12 volunteers (6 male; mean age 23 years [range 21–31 years]) participated. MR data were acquired on a 3T Siemens/Varian MRI System. MRS data was acquired using a 1 channel transmit/receive head-coil from a 2 × 2 × 2 cm voxel centred on the hand-knob area within the motor cortex in the left hemisphere. First, a standard PRESS sequence (TR = 3 s, TE = 68 ms) was used to acquire an unedited spectrum with 32 averages. Then, a MEGA-PRESS sequence (TR = 3 s, TE = 68 ms) was used, with 20 ms double-banded Gaussian inversion pulses for simultaneous spectral editing and water suppression;

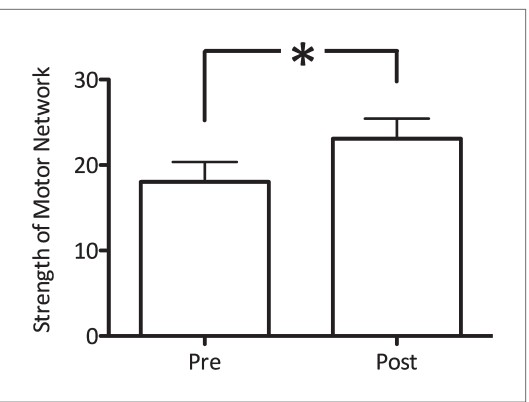

**Figure 4**. Anodal tDCS applied to M1, which is known to decrease GABA levels, significantly increased functional connectivity within the motor RSN (t(9) = 2.59, p=0.02). DOI: 10.7554/eLife.01465.009

the water suppression band was set to a frequency of 4.7 ppm, the editing band alternated between 1.9 ppm and 7.5 ppm, to collect an edited spectrum with 256 averages.

FMRI data were acquired on a separate day using the same scanner as above with a 4 channel receive head coil. 200 axial echo planar volumes were acquired (43 × 3 mm axial slices, TE = 28 ms, TR = 3000 ms, FOV = 192 × 192). Subjects lay at rest with their eyes open for the duration of the FMRI scan, and had performed no tasks prior to this data being acquired.

## Experiment 2

MRS and fMRI data were acquired from 16 volunteers (2 male, mean age 24 years, [range 20–39 years]) on a 3T Siemens Verio in the same session using a 32 channel receive coil.

MRS data were acquired as described in experiment 1, with 288 averages. Data were pre-processed using in-house scripts to combine data from each channel, remove motion corrupted signal averages and correct for frequency drifts during acquisition. 128 axial echo planar volumes of fMRI data were acquired (44 × 3 mm axial slices, TE = 30 ms TR = 2410 ms, FOV = 192 × 100).

## Experiment 3

12 volunteers (6 male, mean age 61 years [range 45–72 years]) participated. MRS data were acquired on a 3T Siemens/Varian MRI System exactly as described in experiment 1. FMRI data were acquired on a 3T Siemens Trio system with a 12 channel receive head coil (60 vol, 35 × 3.8 mm axial slices, TE = 40 ms, TR = 2000 ms, FOV = 192 × 192).

## Experiment 4

10 volunteers (3 male, mean age 22.7 years [range 21–24 years]) participated. FMRI data were acquired exactly as in Experiment 1, before and immediately after tDCS application, which was delivered while subjects lay supine in the MR scanner.

tDCS electrodes (fitted with 5 kΩ resistors; Easycap, GmbH; Germany) were sited prior to entry into the scanner in a standard montage. The active electrode was placed over the left motor cortex, centered 5 cm lateral and 2 cm anterior to Cz. The reference electrode was placed on the right supraorbital ridge. As described previously (*Stagg et al., 2009*), high-chloride EEG paste was used as a conducting medium. Anodal, facilitatory, tDCS was delivered via a DC stimulator (Eldith GmbH; Germany), the current had a ramp-up time of 10 s, was held at 1 mA for 10 min and then ramped down over 10s.

## MR data analysis

### MRS analysis

MRS data analysis was performed using jMRUI v2.2 (http://www.mrui.uab.es/mrui/). Data were smoothed using a 2 Hz Lorentzian filter, and phased with respect to both the 0th and 1st order phase. NAA and Creatine line-widths were obtained from the non-edited PRESS acquisition using a non-linear least square fitting algorithm, and were used to constrain peak fitting for the GABA and glutamate/glutamine resonances. Any spectra with a NAA linewidth of >10Hz were rejected as being of insufficient quality (*Stagg et al., 2009*) (one subject from experiment 1 and three from experiment 3). Poor quality spectra may be caused by a number of factors including excessive motion. All neurotransmitter concentrations are given as a ratio to NAA and corrected for grey matter and white matter concentration as described previously using a T1-weighted high-resolution structural scan acquired immediately prior to the MRS acquisition (*Stagg et al., 2009*).

We chose to use NAA as a reference as we wished to use a reference peak that was acquired simultaneously to our metabolites of interest. Due to our acquisition parameters in experiments 1 and 3, the only consistent reference was NAA. A simultaneously acquired reference peak is highly preferable to control for potential drifts in the spectra during acquisition.

## FMRI analysis for resting state networks

FMRI data processing was carried out using Multivariate Exploratory Linear Optimized Decomposition into Independent Components (MELODIC; version 3.10) part of FSL (FMRIB's Software Library, www.fmrib.ox.ac.uk/fsl) (*Smith et al., 2004*; *Beckmann et al., 2005*; *Jenkinson et al., 2012*). Individual pre-statistics processing consisted of motion correction brain extraction; fieldmap-based EPI unwarping spatial smoothing using a Gaussian kernel of FWHM 6.0 mm and high-pass temporal filtering equivalent to 150.0 s (0.007 Hz). Functional data was aligned to structural images (within-subject) initially using linear registration (FMRIB's Linear Image Registration Tool, FLIRT), then optimized using Boundary-Based Registration approach (*Greve and Fischl, 2009*). Structural images were transformed to standard space using a non-linear registration tool (FNIRT), and the resulting warp fields applied to the functional statistical summary images. Pre-processed functional data for each subject were temporally concatenated across subjects to create a single 4D dataset.

Between-subject analysis was performed using a dual regression technique, as described previously (*Filippini et al., 2009*). Briefly, this approach consisted of three stages. First, the concatenated fMRI data set was decomposed using ICA into 25 components and RSNs of interest were identified using spatial correlations against previously defined maps (*Beckmann et al., 2005*). Next, the dual-regression approach was used to identify the subject-specific RSN maps. This process involved using all 25 components to perform a spatial regression against each separate fMRI data set and then using the resulting normalized time-course matrices to perform a temporal regression to estimate subject-specific maps that reflect the subject specific strength of functional connectivity. The resulting subject-specific RSN map was then masked by the group mean RSN map and the mean value within this region extracted for each subject, used as a measure of the strength of functional connectivity within the RSN. Motion is known to have complex effects on resting state connectivity on a subject-by-subject basis (*Power et al., 2012*), but effects of head motion are greatly ameliorated by the dual regression approach used here, compared against simpler single-regression-based analyses (*Zuo et al., 2010*).

## FMRI analysis for seed-based correlations

Data were preprocessed and registered to standard space as described above. For Experiments 1–3, the MRS voxel of interest each subject was registered to standard space to give a region of interest (ROI) for the left M1, and then flipped about the midline to give a ROI for the right M1. An ROI of the PMd was derived from a connectivity-based parcellation of the lateral premotor cortex (*Tomassini et al., 2007*), and was masked by the group mean RSN mask.

The timecourse of the fMRI signal from these ROIs was then extracted from the fMRI data, pre-processed as described above. For each subject left and right M1s and left M1 and left PMd were correlated using Pearson's correlation coefficient. As Pearson's correlation coefficients are non-normally distributed, the r value for each subject was then log-transformed to give a parametric measure of the degree of functional correlation between the ROIs. This was then correlated for each subject with GABA levels acquired from the left M1.

For Experiment 4, where no MRS was acquired, the mean MRS voxel location from all subjects in Experiments 1–3 was calculated and this was used as the ROI.

## Acknowledgements

We are grateful to Dr Jakob Blicher and Dr Manus Donahue for assistance with data collection and to Dr Christian Beckmann for advice on data analysis. This study was supported by the National Institute for Health Research (NIHR) Oxford Biomedical Research Centre based at Oxford University Hospitals NHS Trust and University of Oxford [CJS & HJB]. CJS holds a Sir Henry Dale Fellowship jointly funded by the Wellcome Trust and the Royal Society (Grant Number 102584/Z/13/Z). HJB is a Wellcome Trust Senior Research Fellow.

# Additional information

### Funding

| Funder | Author |
| --- | --- |
| The Wellcome Trust | Heidi Johansen-Berg |

| Funder | Author |
|---|---|
| National Institute for Health Biomedical Research Centre based at the Oxford University Hospitals NHS Trust | Charlotte J Stagg, Heidi Johansen-Berg |

The funders had no role in study design, data collection and interpretation, or the decision to submit the work for publication. The views expressed are those of the authors and not necessarily those of the NHS, the NIHR or the Department of Health.

## Author contributions

CJS, Conception and design, Acquisition of data, Analysis and interpretation of data, Drafting or revising the article; VB, UA, CAG, ASI, Acquisition of data, Drafting or revising the article; CS-B, MW, SMS, NF, JN, Analysis and interpretation of data, Drafting or revising the article; JO'S, Acquisition of data, Analysis and interpretation of data, Drafting or revising the article; HJ-B, Conception and design, Analysis and interpretation of data, Drafting or revising the article

## Ethics

Human subjects: Informed consent, including consent to publish was obtained from all subjects. The study was performed in accordance with local ethics committee approval (Experiments 1, 3 & 4: Oxfordshire REC A 06/Q1604/2; Experiment 2: East London REC 1 10/H0703/50).

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
