## [Decision Letter]

Thank you for sending your work entitled “Local GABA concentration is related to network-level resting functional connectivity” for consideration at *eLife*. Your article has been favorably evaluated by a Senior editor and 3 reviewers, one of whom is a member of our Board of Reviewing Editors.

The following individuals responsible for the peer review of your submission have agreed to reveal their identity: Jody Culham, Reviewing editor; Peter Fransson, peer reviewer.

The Reviewing editor and reviewers discussed their comments before we reached this decision, and the Reviewing editor has assembled the following comments to help you prepare a revised submission.

Two external reviewers and the Reviewing editor all provided quite favorable reviews of the manuscript. The consensus was that the experiments were well-conceived and well-executed and the manuscript was well-written. All three reviewers also thought that the findings make an important contribution to the literature, especially given the ubiquity of fMRI studies using resting state correlations and the field's very limited understanding of the neuronal basis of these fluctuations in the BOLD signal.

Several issues need to be addressed in a moderate revision:

1) Impact of motion – as shown in recent papers by Petersen's group and others motion can have a large and hard to predict impact on resting state BOLD correlation maps. Similarly motion can degrade the quality of the GABA measurement. It would be good if the authors added in the text a paragraph about how motion was addressed for both measurements. For the resting state BOLD measurements were any of the advanced motion corrections like the 'scrubbing' method proposed by Petersen and colleagues used?

2) Spectral quality – the spectral quality of the previous work by the Oxford group has been excellent. However it would still be useful to provide as supplementary material, or at least to the reviewers, a GABA spectrum from a subject with a low and high GABA/NAA ratio since the paper's main conclusion depends on this comparison.

3) NAA as a reference – the authors should provide a short explanation of why this is preferred over creatine or an absolute water reference.

4) GABA tone and cellular GABA concentration – the authors should expand on this relationship to make clear to readers not familiar with MRS.

5) A strength is the demonstration of the specificity of the correlation between GABA and interhemispheric M1-M1 connectivity (and M1-GABA does not predict connectivity within another network, the DMN). However, it would also be quite valuable to know whether the GABA concentrations are related to the connections between M1 and other areas. That is, the interhemispheric connections between homotopic regions (especially motor cortex) may rely more on mutual inhibition than connections within a hemisphere. I would like to see an investigation of the correlation between M1 and another region, perhaps a secondary motor area known to be connected with M1 (perhaps dorsal or ventral premotor cortex).

6) The paper includes a lot of replication. On the one hand, this is a good thing. Psychology and neuroscience have recently come under fire for frequent failures to replicate and so it is reassuring to see an effect that is replicated twice and replicated across approaches (ICA and seed-based RSC analyses), across data collection schemes (fMRI and MRS in separate vs. the same session), across scanner configurations (two Siemens scanners with different coils) and across populations (including both the young and the elderly). It would be worth emphasizing the replicability across all these factors more in the Discussion. On the other hand, the replication now interferes to some degree the flow of the paper. Can the authors take advantage of some of the on-line features of embedding supplemental figures or make a table or flow diagram that can mitigate this?

---

## [Author Response]

*1) Impact of motion – as shown in recent papers by Petersen's group and others motion can have a large and hard to predict impact on resting state BOLD correlation maps. Similarly motion can degrade the quality of the GABA measurement. It would be good if the authors added in the text a paragraph about how motion was addressed for both measurements. For the resting state BOLD measurements were any of the advanced motion corrections like the 'scrubbing' method proposed by Petersen and colleagues used*?

We recognize that motion can cause artifacts in both resting BOLD and GABA MRS measurements. Many different approaches exist to deal with this problem and we believe that the approaches we have taken (detailed below) will have dealt adequately with motion. However, no approach to motion correction is perfect; it is therefore important also to note that, while common variation in data quality (e.g., due to head motion) could theoretically contribute to correlation between signals, we cannot envisage a mechanism by which the specificity of the relationships we report could be explained by such factors.

For our BOLD resting state analyses, we would first like to reassure the editor and the reviewers that all our data were of high quality; we have extracted the mean movement from each of the RSN scans in terms of relative displacement in mm detected by motion correction. For the 4 experiments included in this study, the mean displacement over the course of the RSN scan was consistently low: Experiment 1: 0.06±0.007mm; Experiment 2: 0.08±0.02mm; Experiment 3: 0.04±0.01mm; Experiment 4 Pre: 0.05±0.01mm; Post 0.07±0.02mm.

However, the issue of motion is not trivial, even in high quality data. As the reviewers and editor suggest, a number of different potential approaches exist to deal with this issue. In this paper we have chosen to use an ICA-based analysis, with network strength quantified for each subject using a dual regression analysis. Because of the nature of the applied two multiple regressions (one spatial and one temporal), in general, effects of head motion are greatly ameliorated, compared against simpler single-regression-based analyses (25).

We have clarified this in the manuscript (revisions in italics):

“The resulting subject-specific RSN map was then masked by the group mean RSN map and the mean value within this region extracted for each subject, used as a measure of the strength of functional connectivity within the RSN. *Motion is known to have complex effects on resting state connectivity on a subject-by-subject basis* (15), *but effects of head motion are greatly ameliorated by the dual regression approach used here, compared against simpler single-regression-based analyses (25)*.”

In terms of spectral quality, motion would impact on spectral quality primarily as an increase in linewidth. We have stringent a priori spectral quality control, which ensures that spectra that were not of adequate quality were excluded from the analysis. Any spectra with a linewidth > 10Hz were excluded from analysis, a standard objective criteria of spectral quality.

We have added the following to the results section to clarify this (revisions in italics):

“Any spectra with a NAA linewidth of > 10Hz were rejected as being of insufficient quality (20) (1 subject from experiment 1 and 3 from experiment 3). *Poor quality spectra may be caused by a number of factors including excessive motion.*”

*2) Spectral quality – the spectral quality of the previous work by the Oxford group has been excellent. However it would still be useful to provide as supplementary material, or at least to the reviewers, a GABA spectrum from a subject with a low and high GABA/NAA ratio since the paper's main conclusion depends on this comparison*.

We are happy to provide these data as a supplemental figure. We have added a reference to this in the manuscript to point the reader to these data.

3) NAA as a reference – the authors should provide a short explanation of why this is preferred over creatine or an absolute water reference.

For all spectra it is desirable to reference the metabolites of interest to a simultaneously acquired reference peak. Due to our acquisition parameters in Experiments 1 & 3, the only simultaneously acquired peak available was NAA, and therefore we used this throughout for consistency. The use of a simultaneously acquired reference is vital to account for any possible signal drifts over the acquisition of the spectra.

We did not acquire an external water reference for any of the experiments and therefore do not believe that an absolute water reference would be more accurate than a ratio to NAA, especially as this would not be a simultaneously acquired reference.

It is theoretically possible by referencing our metabolites of interest to NAA, that the relationships described in the manuscript between GABA levels and network strength might be driven by differences in NAA (a marker of neuronal mitochondrial activity) rather than GABA. This is highly unlikely as NAA is used as the reference for Glx as well, and therefore the specificity of the relationship between network strength and GABA speaks to this being the driver of this relationship.

However, to ensure that this was not the case, we performed the analyses in the paper using a creatine reference. A simultaneously acquired creatine reference peak was acquired for Experiment 2. For experiments 1 & 3 we acquired a PRESS spectrum with a 68ms TE before the GABA measurement, from which it is possible to acquire a creatine peak as a reference. We also ran the relevant analyses using this (non-simultaneous) creatine peak and are pleased to say that the pattern of results reported in the manuscript as referenced to NAA holds for this analysis referenced to creatine.

We would prefer to keep NAA as our reference peak, for reasons outlined above, and therefore have not altered the data in the manuscript. We are happy to do so, however, if the editor and the reviewers feel that it is appropriate.

We have added the following to the Discussion section to highlight our reasons for our choice of reference peak:

“We chose to use NAA as a reference as we wished to use a reference peak that was acquired simultaneously to our metabolites of interest. Due to our acquisition parameters in experiments 1 & 3, the only consistent reference was NAA. A simultaneously acquired reference peak is highly preferable to control for potential drifts in the spectra during acquisition.”

*4) GABA tone and cellular GABA concentration – the authors should expand on this relationship to make clear to readers not familiar with MRS*.

We are happy to expand this point and have added the following to the manuscript:

“Evoked gamma power is also known to be inversely related to extra-synaptic GABA tone (23). GABA is present in the human brain in three major pools – in pre-synaptic vesicles, as a metabolite in the cytoplasm, and in the extracellular fluid, where it underlies extra-synaptic GABA tone. It is likely that MRS-assessed GABA, which is sensitive to the total *amount* of GABA within the voxel, more closely reflects extra-synaptic rather than synaptic GABA activity (21).”

*5) A strength is the demonstration of the specificity of the correlation between GABA and interhemispheric M1-M1 connectivity (and M1-GABA does not predict connectivity within another network, the DMN). However, it would also be quite valuable to know whether the GABA concentrations are related to the connections between M1 and other areas. That is, the interhemispheric connections between homotopic regions (especially motor cortex) may rely more on mutual inhibition than connections within a hemisphere. I would like to see an investigation of the correlation between M1 and another region, perhaps a secondary motor area known to be connected with M1 (perhaps dorsal or ventral premotor cortex)*.

We chose to look at the connections between the two M1s as the major cortical connectivity from M1 is to the homotopic region. However, it is true that it would be interesting to investigate the relationship between M1 and another cortical region. At the reviewers’ suggestion we chose to investigate connectivity between the M1 and the ipsilateral PMd, and found a trend for a similar effect.

*6) The paper includes a lot of replication. On the one hand, this is a good thing. Psychology and neuroscience have recently come under fire for frequent failures to replicate and so it is reassuring to see an effect that is replicated twice and replicated across approaches (ICA and seed-based RSC analyses), across data collection schemes (fMRI and MRS in separate vs. the same session), across scanner configurations (two Siemens scanners with different coils) and across populations (including both the young and the elderly). It would be worth emphasizing the replicability across all these factors more in the Discussion. On the other hand, the replication now interferes to some degree the flow of the paper. Can the authors take advantage of some of the on-line features of embedding supplemental figures or make a table or flow diagram that can mitigate this*?

In order to improve the flow of the paper we have moved experiments 2 & 3 to the supplementary data, keeping only the major findings in the main paper. We believe that this has substantially improved the paper. In addition, we are happy to highlight the strength of the replication within the study as suggested and have added the following to the Discussion (revisions in italics):

“In this study we investigated the basis of the long-range fluctuations seen in resting fMRI. In three separate cohorts of subjects we have demonstrated a significant negative correlation between the levels of GABA in M1 and the strength of functional connectivity within the motor RSN. This relationship was specific both anatomically and neurochemically: *no relationship between glutamate levels and resting connectivity was demonstrated. We believe the fundamental relationship between resting connectivity within the motor network and M1 GABA demonstrated here is robust, as it is replicated across both ICA and seed-based approaches; whether the MRS and fMRI data were collected on the same day or different days; across different scanners; and in young and old healthy cohorts.*”